# A Multi-Scale Temporal Convolutional Network with Attention Mechanism for Force Level Classification during Motor Imagery of Unilateral Upper-Limb Movements

**DOI:** 10.3390/e25030464

**Published:** 2023-03-07

**Authors:** Junpeng Sheng, Jialin Xu, Han Li, Zhen Liu, Huilin Zhou, Yimeng You, Tao Song, Guokun Zuo

**Affiliations:** 1Faculty of Information Science and Technology, Ningbo University, Ningbo 315211, China; 2Cixi Institute of Biomedical Engineering, Ningbo Institute of Materials Technology and Engineering, Chinese Academy of Sciences, Ningbo 315300, China; 3University of Chinese Academy of Sciences, Beijing 100049, China

**Keywords:** motor imagery (MI), unilateral upper-limb dynamic state, force, electroencephalograph (EEG), multi-scale temporal convolutional network (MSTCN), attention mechanism

## Abstract

In motor imagery (MI) brain–computer interface (BCI) research, some researchers have designed MI paradigms of force under a unilateral upper-limb static state. It is difficult to apply these paradigms to the dynamic force interaction process between the robot and the patient in a brain-controlled rehabilitation robot system, which needs to induce thinking states of the patient’s demand for assistance. Therefore, in our research, according to the movement of wiping the table in human daily life, we designed a three-level-force MI paradigm under a unilateral upper-limb dynamic state. Based on the event-related de-synchronization (ERD) feature analysis of the electroencephalography (EEG) signals generated by the brain’s force change motor imagination, we proposed a multi-scale temporal convolutional network with attention mechanism (MSTCN-AM) algorithm to recognize ERD features of MI-EEG signals. Aiming at the slight feature differences of single-trial MI-EEG signals among different levels of force, the MSTCN module was designed to extract fine-grained features of different dimensions in the time–frequency domain. The spatial convolution module was then used to learn the area differences of space domain features. Finally, the attention mechanism dynamically weighted the time–frequency–space domain features to improve the algorithm’s sensitivity. The results showed that the accuracy of the algorithm was 86.4 ± 14.0% for the three-level-force MI-EEG data collected experimentally. Compared with the baseline algorithms (OVR-CSP+SVM (77.6 ± 14.5%), Deep ConvNet (75.3 ± 12.3%), Shallow ConvNet (77.6 ± 11.8%), EEGNet (82.3 ± 13.8%), and SCNN-BiLSTM (69.1 ± 16.8%)), our algorithm had higher classification accuracy with significant differences and better fitting performance.

## 1. Introduction

Motor imagery (MI) is a brain–computer interface (BCI) paradigm in conformity with the normal activity of the human mind [1,2]. MI induces a decrease or increase in power in the alpha (8–14 Hz) and beta (15–30 Hz) bands for electroencephalograph (EEG) signals, which is an event-related de-synchronization/synchronization (ERD/ERS) phenomenon [3].

In MI-based BCI research, a rehabilitation robot has been used as an external device to constitute a brain-controlled rehabilitation robot system [4,5,6,7,8]. The system can create a new artificial neural pathway between the brain and the affected limb using closed-loop feedback control, allowing patients who have suffered a brain injury (e.g., stroke) to achieve neuroplasticity and improve their limb movement function. As a result, the system has greater clinical application potential. Current MI-BCI employs the MI paradigms of simple limb movement (e.g., left hand, right hand, or feet) generally [9,10,11,12,13]. These MI paradigms induce a small number of ERD/ERS patterns and provide limited commands to control the movement directions of the rehabilitation robot. However, in a brain-controlled rehabilitation robot system, robot-assisted training mainly emphasizes the dynamic process of force interaction between the robot and patient. Moreover, due to the patient’s muscle strength level and movement ability, the robot needs to provide assisted force according to the patient’s demand to help the patient complete limb movement training tasks. Therefore, it is critical for clinical application of a brain-controlled rehabilitation robot system to induce and recognize the thinking states of patients’ demand for assistance and allow their brains to control the rehabilitation robot with force in the movement direction.

In recent years, some researchers have investigated the linear correlation between the ERD phenomenon and the kinetic parameter of limb force [14,15,16]. They studied the ERD feature recognition of EEG signals generated by the brain’s force change motor imagination, which was thus used for BCI control. Xu et al. [14] designed a right-hand clenching MI paradigm in which three levels of force (20%, 50%, and 80% maximum voluntary contraction (MVC)) were introduced. They used the Hilbert–Huang transform (HHT) and support vector machine (SVM) algorithms for six subjects’ MI-EEG data to extract ERD features and classify them, with an average classification accuracy of 92%. Wang et al. [15] designed a right-hand clenching MI paradigm in which three levels of force (0%, 10%, and 30% MVC) were introduced. They used the multi-class common spatial pattern (CSP) and SVM algorithms for six subjects’ MI-EEG data to extract ERD features and classify them, with an average classification accuracy of 71%. After that, Wang et al. [16] used the CSP algorithm to extract ERD features from nine subjects’ EEG data of right-hand clenching imagination (force levels of 10% and 30% MVC). They also used the fourth-order Butterworth bandpass filter to extract movement-related cortical potentials (MRCPs) features in the 0.3–3 Hz frequency band. Finally, the SVM algorithm was used to classify the combined features, with an average classification accuracy of 78.3%.

The above studies can extend the number of recognizable thinking categories in MI-BCI through limb force. However, MI paradigms of force under a unilateral upper-limb static state are designed, which are difficult to be applied to the dynamic force interaction process between the robot and patient in the brain-controlled rehabilitation robot system. The process requires induction of the patient’s thinking states of assisted force. For MI-EEG feature recognition in these studies, traditional methods rely on manual feature extraction, whose feature learning is insufficient. In particular, when the number of thinking categories increases, classification accuracy is low. Therefore, some researchers have begun to conduct research on deep learning algorithms for MI-EEG feature recognition. Yang et al. [17] used the CSP algorithm to extract space domain features of MI-EEG signals and then used a convolutional neural network (CNN) to learn deep-level features. However, due to the poor learning of EEG temporal features by CNN, the algorithm did not learn features adequately and performed poorly in classification. Ma et al. [18] proposed a time-distributed attention (TD-Atten) algorithm for MI-EEG feature recognition. A sliding window approach was used in this algorithm to slice EEG signals continuously, and then a one-versus-rest filter bank common spatial pattern (OVR-FBCSP) algorithm was used to extract frequency–space domain features from the sliced EEG signals. Finally, the attention mechanism and long short-term memory (LSTM) network were used to further extract EEG temporal features. However, the TD-Atten algorithm still extracted EEG features manually, and the classification accuracy was limited by the quality of feature extraction. Schirrmeister et al. [19] proposed the end-to-end algorithm to learn features from MI-EEG data automatically, which avoided manual feature extraction. Because of the strong time correlation of EEG signals, a temporal convolutional network (TCN) was mainly used in this end-to-end algorithm to extract the features of EEG over time to complete the classification. However, a single-scale convolution kernel was only used in TCN to extract EEG features, and the feature information was relatively simple [20].

Therefore, in this paper, we designed a MI paradigm of force under a unilateral upper-limb dynamic state and analyzed the MI-EEG signals’ ERD features in nine subjects. In particular, a multi-scale temporal convolutional network with attention mechanism (MSTCN-AM) algorithm was proposed to recognize ERD features of MI-EEG signals. In this algorithm, the multi-scale temporal convolutional network (MSTCN) module was designed to simultaneously learn multi-dimensional time–frequency domain features of EEG signals. The problem of limited features in single-scale convolution kernel learning was avoided. Furthermore, to solve the problem of feature redundancy, the time–frequency–space domain features obtained by the MSTCN and spatial convolution modules were dynamically weighted by the attention mechanism. The attention mechanism was designed to learn the input features and obtain the attention weight dynamically so as to help the algorithm focus on the features related to MI thinking categories with three-level force. This research will lay the foundation for force control of a brain-controlled rehabilitation robot system in the movement direction.

## 2. Materials

### 2.1. Participants

Data were collected from 12 healthy subjects (six males and six females, age from 22 to 26 years). All subjects were right-handed and had no history of psychiatric disorders or limb movement disorders. During EEG signal preprocessing, data from three subjects were discarded because of poor EEG data quality (e.g., strong drifts or extensive muscle artifacts). As a result, the EEG data of nine subjects (S1–S9) were retained. The experiment was approved by the Bioethics Committee at Cixi Institute of Biomedical Engineering and carried out at the institute. Before the experiment, all subjects were informed of the experimental procedure and signed an informed consent form. The subjects were given a one-week training session to familiarize themselves with the experimental procedure. The training session was divided into two stages composed of a motor execution stage and a motor imagery stage [21]. During the MI process, subjects were asked to do their best to imagine their limb movements so as to maximize the activation of their brain hemispheres.

### 2.2. Experimental Procedure

In this paper, the MI paradigm of force under a unilateral upper-limb dynamic state was designed based on the movement of wiping a table in human daily life (the wiping direction was an alternating front and back cycle). For this paradigm, we introduced three levels of force (small, medium, and large force) into the same dynamic movement of the right upper limb, and the corresponding MI scenes were wiping dust, dirt, and tea stains, respectively. The force levels in this paradigm were designed according to the difficulty of wiping the table, which facilitated better imagination by the subjects. During a round of the MI experiment, the force level of imagination was consistent. The random switching of force levels among single trials would interfere with the thinking states of subjects, thus accelerating their brain tiredness. Each subject was required to perform nine rounds of the MI experiment, and each round had 24 single trials. In the nine experimental rounds, every three rounds were arranged in ascending order of force levels. At the end of each round of the experiment, subjects had a rest of 5–10 min.

The single trial procedure is shown in Figure 1. It was divided into 4 periods with a total time of 16 s. The first period was the preparation part, in which a white circle appeared in the center of the computer screen for 2 s and the subject remained relaxed. The second period was the cue part, in which the white circle disappeared and a text cue of MI appeared on the screen. It lasted for 4 s and the subject remained relaxed with no movement output. The third period was the MI part, in which the black screen lasted for 6 s and the subject performed the MI procedure. The fourth period was the rest part, in which a text cue of rest appeared on the screen for 4 s and the subject remained relaxed.

### 2.3. Data Collection and Preprocessing

We used a Neuroscan EEG system with a sampling rate of 1000 Hz to collect EEG signals. An electrode cap with 32 Ag/AgCl electrodes was used, and the distribution of electrodes followed the international 10/20 system. During the experiment, the impedance between the electrode and scalp was less than 10 kΩ.

For EEG signal preprocessing, we first used the re-referencing method of bilateral mastoid referencing, and a 0.5–100 Hz band-pass filter and a 50 Hz notch filter were used. Furthermore, we used the common average reference (CAR) method to improve the spatial signal-to-noise ratio of EEG signals, and the finite impulse response (FIR) filter was also used to filter EEG signals from 8–30 Hz to remove movement artifacts. Then, the EEG signals were down-sampled to 128 Hz. In addition, the “HEOG” and “VEOG” electrode signals were removed in this preprocessing process.

## 3. Methods

### 3.1. MI-EEG Feature Analysis

Event-related spectral perturbation (ERSP) is a method to characterize time–frequency domain features of single-channel EEG signals. It can reflect variation in the power spectrum of EEG signals relative to the baseline [18,22]. This variation can reflect the ERD/ERS features of MI-EEG signals [23]. Therefore, the ERSP method was used in this paper to perform a time–frequency domain analysis of ERD features of MI-EEG signals among different levels of force. The formula is as follows:(1)$$ERSP\left(f,t\right)=\frac{1}{n}\sum_{k=1}^{n}\left(F_{k}{\left(f,t\right)}^{2}\right)$$
where *n* denotes the number of trials and $F_{k}\left(f,t\right)$ denotes the spectral estimate of the *k*th trial MI-EEG signals at frequency *f* and time *t*. We used short-time Fourier transform (STFT) to calculate the ERSP value (dB). For each channel, we calculated mean ERSP values of MI-EEG signals of nine subjects at each force level in the frequency range of 8–30 Hz and the time range of −1–6 s. The start of the MI period was taken as 0 s, and the data of one second before the MI period (−1–0 s) were taken as the baseline data. Based on the above calculated ERSP values in time–frequency domain analysis, we also calculated mean ERSP values of MI-EEG signals of nine subjects at each force level in two frequency bands (alpha and beta bands) and the time range of 0–6 s to analyze space domain features for all channels, which were shown by brain topography maps.

### 3.2. MI-EEG Feature Recognition Algorithm

In this paper, we proposed the MSTCN-AM algorithm to recognize ERD features of MI-EEG signals. The network structure is shown in Figure 2, and the network structure parameters are shown in Table 1. In this algorithm, the MSTCN module, spatial convolution module, and attention module were mainly designed to learn EEG features progressively. Then, the fully connected layer was used to complete the classification. 

#### 3.2.1. MSTCN Module

EEG is a one-dimensional, time-varying, and non-stationary random signal with a small amount of information and a low signal-to-noise ratio. In this paper, the MI paradigm was designed to induce subjects to imagine different levels of force for the same dynamic movement of their right upper limb. Through time–frequency domain analysis, we found that the feature frequency bands of MI-EEG signals at each force level overlapped. The ERD features in the feature frequency bands became more apparent as the force level of imagination increased. However, the feature differences were slight among single-trial MI-EEG signals at different levels of force. Although the single-scale TCN commonly used could learn time–frequency domain features of EEG signals [19,24], it could not extract the slight features of single-trial EEG signals sufficiently [20].

Therefore, we designed the MSTCN module with three small-size convolution kernels. The module could extract multi-dimensional fine-grain time–frequency domain features from the preprocessed single-trial MI-EEG signals. There were certain size differences among adjacent convolution kernels, which could learn the features of different frequencies. The obtained feature information was sufficient, which enabled the algorithm to classify EEG signals with slight feature differences. The module structure is shown in Figure 3, and the module structure parameters are shown in Table 1. The feature input of this module is defined as X∈ℝ1×N×M*,* which has three dimensions (channel, height, and width), where *N* and *M* are the number of electrodes and time points of EEG data, respectively. It was obtained by the dimensional transformation of raw EEG signals in the reshape layer. We define the sizes of three convolution kernels as (1, S) in the module. The equation is as follows:(2)Xi=Bi(Convi(X)), i=1⋯3
where Bi(·) represents the batch normalization (BN) layer [25]. The enhanced EEG time–frequency domain feature information Xc=[X1,X2,X3], Xc∈ℝ24×N×M was obtained by concatenating the multi-dimensional features extracted from three small-size convolution kernels. Meanwhile, this multi-scale convolution processing could filter EEG signals to eliminate the noise influence and increase the robustness of the algorithm.

#### 3.2.2. Spatial Convolution Module

In the space domain analysis of different-level-force MI-EEG signals, we found that the activation area of the brain’s contralateral sensorimotor area overlapped. However, the activation area became wider as the force level of imagination increased. In other words, the space features of EEG signals were different around the feature overlap area. Therefore, we used the spatial convolution module to further learn space domain features of EEG signals. Then, the average pooling layer was connected after the spatial convolution module. The pooling kernel with a large size was used to aggregate the extracted EEG time–frequency–space domain features. Therefore, information redundancy in the features’ width dimension was reduced to avoid over-fitting of the algorithm. The size of the spatial convolution kernel is (N, 1), where *N* is the number of electrodes of EEG data. The size of the pooling kernel is (1, K). The formula is as follows:(3)$$X_{s}=H\left(Conv\left(X_{c}\right)\right)$$
we define H(·) as a composite layer of four consecutive processing units: BN, the exponential linear unit (ELU) activation function [26], the average pooling layer, and the dropout layer [27]. The parameters information of this module is shown in Table 1.

#### 3.2.3. Attention Module

The attention mechanism originates from the research of human vision. By selectively focusing on a portion of features while ignoring others [28,29,30], it can help algorithms improve their sensitivity to features and classification performance. Therefore, for time–frequency–space domain features extracted by the MSTCN and spatial convolution modules, we designed the attention module using channel attention mechanism [31] to focus on the channel features related to MI thinking categories with three-level force. It adaptively adjusted the feature response among feature channels through recalibration to obtain the weighted features, and its function was to reduce information redundancy in the features’ channel dimension. The module structure diagram is shown in Figure 4. Assuming that Xs∈ℝC×H×W is the feature input of the attention module, the height and width information of Xs are aggregated by the average pooling and max pooling methods to work out the average pooling and max pooling features as Xavgc and Xmaxc. After that, two feature maps are obtained by passing Xavgc and Xmaxc through a shared network. In order to generate the final channel attention weight Mc∈ℝC×1×1, the two feature maps are processed using the element-wise summation and sigmoid function. The shared network is a multi-layer perceptron (MLP) model with one hidden layer, and the size of the hidden layer is set to ℝC/r×1×1, where *r* represents the reduction ratio. The formula is as follows:(4)Mc(Xs)=σ(MLP(AvgPool(Xs))+MLP(MaxPool(Xs)))=σ(W1(W0(Xavgc))+W1(W0(Xmaxc)))
where W0∈ℝC/r×H and W1∈ℝC×C/r are the parameter matrices of the MLP model. The MLP model continuously learns and updates its parameter matrices through the gradient descent method, and the cross-entropy loss function is used as the objective function of the gradient descent method. In the process of minimizing the loss function, it made the parameter optimization more stable by using the logarithmic function. Therefore, the optimized attention module could better build the correlation among features of different feature channels, thus helping the algorithm better fit EEG data and improve its classification accuracy. The cross-entropy loss value is calculated as follows:(5)Loss=−∑kYklog(Pk)
where Yk and Pk are the probabilities of the true category and prediction category of *k*th trial MI-EEG signals, respectively. The learning process of parameter matrices is as follows:(6)Wijnew=Wijold−lr·∇WL(Wijold)
where Wijnew and Wijold are the updated value and original value of the parameter matrices, *lr*, L(·) and ∇WL(Wijold) are the learning rate, cross-entropy loss function, and gradient of the loss function at Wijold value, respectively. Finally, the feature output of the attention module is Xf∈ℝC×H×W. The formula is as follows:(7)Xf=Mc(Xs) ⨂ Xs

## 4. Experiments and Results

### 4.1. MI-EEG Feature Analysis Results

After time–frequency analysis, we found that the feature differences were slight among single-trial MI-EEG signals at different levels of force. Therefore, for MI-EEG signals of all subjects at each force level, we plotted mean time–frequency maps, mean power brain topography maps, and ERSP curves, as shown in Figure 5.

The mean time–frequency maps at C3 and C4 channels are shown in Figure 5a. The ERD features appear in both the alpha and beta bands, and the feature frequency bands of three-level-force MI-EEG signals overlap. However, the ERD features in the feature frequency bands become more apparent as the force level of imagination increases. The above ERD features mainly appear at 13–14 Hz in the alpha band and 24–28 Hz in the beta band. Therefore, we calculated mean ERSP values of these two frequency bands for all channels to plot mean power brain topography maps, as shown in Figure 5b. The brain topography maps show that the brain’s contralateral sensorimotor area is activated, which corresponds to three-level-force MI under the right upper-limb dynamic state, and, as the force level of imagination increases, the activation area becomes wider. The space features of MI-EEG signals are different around the feature overlap area.

For C3 channel, we also calculated average power based on ERSP values to plot ERSP curves in two frequency bands (13–14 Hz and 24–28 Hz), as shown in Figure 5c,d. As we can see, the power in two frequency bands of three-level-force MI-EEG signals gradually decreases with time, and it decreases more with time as the force level increases.

### 4.2. Other MI-EEG Feature Recognition Algorithms

In order to verify the superiority of the MSTCN-AM algorithm on feature recognition of three-level-force MI-EEG signals in this paper, some excellent algorithms in current EEG feature recognition research were selected as baseline algorithms for comparison. Baseline algorithms are described as follows:

OVR-CSP+SVM [32,33]: CSP is a space domain feature extraction algorithm for EEG signals in binary classification tasks. The matrix diagonalization method is used to find a set of optimal spatial filters to project EEG signals so that the variance difference between two classes of signals is maximized. The OVR-CSP algorithm is an extended form of CSP. It is used for feature extraction of EEG signals in multi-classification tasks, and SVM is a machine learning algorithm for feature classification of EEG signals.

Deep ConvNet and Shallow ConvNet [19]: The TCN is used in both algorithms to extract time–frequency domain features of EEG signals. The spatial convolution layer is used to extract space domain features, and the fully connected layer is used to complete classification. The difference between the two algorithms is deep-level feature extraction. For Deep ConvNet, the max pooling layer is used to reduce the dimension of the feature map. Then, it is combined with multiple standard convolution layers to further extract deep-level features. However, the square nonlinear activation function, average pooling layer, and logarithmic activation function are used in Shallow ConvNet to process the feature input to extract deep-level features.

EEGNet [24]: This is a compact CNN algorithm that progressively extracts EEG signal features using multiple convolution layers. The separable convolutional network is applied, which contains the depthwise convolution and pointwise convolution [34], thus significantly reducing the training parameters in the network model. Finally, classification is completed through the fully connected layer.

SCNN-BiLSTM [35]: This is a hybrid deep learning algorithm. The shallow convolutional neural network (SCNN) is used to extract the frequency–space domain features of EEG signals, and then the BiLSTM network is followed to extract the time domain features. What is more, the attention mechanism is adopted to weight time–frequency–space domain features dynamically. Finally, classification is completed through the fully connected layer.

In order to make the training of the above network models more stable, we normalized the preprocessed EEG data X∈ℝN×M. The normalization equation is:(8)x*=x−μσ
where *N*, *M* represent the number of electrodes and the time points, *μ* and *σ* are mean and standard deviation of EEG data, respectively. Accuracy as the evaluation indicator to assess the results of algorithms is calculated as follows:(9)accuracy=TP+TNTP+TN+FP+FN
where *TP* indicates that the true category and prediction category of EEG signals are both positive. *TN* indicates that the true category and prediction category of EEG signals are both negative. *FP* indicates that the true category of EEG signals is negative but the prediction category is positive. *FN* indicates that the true category of EEG signals is positive but the prediction category is negative.

### 4.3. MI-EEG Feature Recognition Results

The hardware resources we used were an Intel Core i7-10700F @2.90GHz CPU, a Nvidia RTX 2060 GPU, and 80GB RAM. The network model of the MSTCN-AM algorithm was based on the PyTorch deep learning framework [36]. It was trained in a supervised manner. During the training process of the algorithm, we chose the cross-entropy loss function. The number of iterations was 500. To accelerate the convergence speed of the algorithm, we chose the Adam optimizer [37]. The batch size and learning rate were 16 and 0.001, respectively.

We used four-fold cross-validation to measure the generalization ability of algorithms. The results (accuracy (%)) are presented in Table 2 as *mean ± standard deviation (std)*. The accuracy of the MSTCN-AM algorithm is 86.4 ± 14.0%, which is higher than that of other baseline algorithms. However, in all algorithms, the accuracy of subject S2 is substantially lower than that of other subjects. In addition, we used paired *t*-test to analyze whether there was a significant difference in accuracy between the MSTCN-AM algorithm and other baseline algorithms. Statistical analysis results are shown in Figure 6. The classification accuracy of the MSTCN-AM algorithm is significantly different from that of other baseline algorithms (*p* < 0.05). Furthermore, we plotted confusion matrices of MI-EEG data test sets for all subjects, as shown in Figure 7. In most subjects, the MSTCN-AM algorithm has a high classification accuracy for EEG signals at each force level, but the accuracy of subject S2 is low.

Among the baseline algorithms, the classification accuracy of the EEGNet algorithm is 82.3 ± 13.8%, which is only lower than that of the MSTCN-AM algorithm. Therefore, we trained their network models and further compared their fitting performance for MI-EEG data. The loss value curves and accuracy curves of three subjects’ (S5, S6, and S7) MI-EEG data test sets are shown in Figure 8. During the iteration of network models, the loss values of two algorithms show a rapid decrease toward convergence. However, the loss values of the EEGNet algorithm show a short-term increase in the early part of the iteration. In addition, the accuracy curves show that the MSTCN-AM algorithm has a higher convergence rate than the EEGNet algorithm.

### 4.4. The Effect of Size Differences of Adjacent Convolution Kernels on the MSTCN-AM Algorithm’s Classification Result

In the MSTCN module of the MSTCN-AM algorithm, three small-size convolution kernels were used to extract fine-grained time–frequency domain features of different dimensions from MI-EEG signals, and the combination of different-size convolution kernels determined the difference in the structure of the MSTCN module. Therefore, a combination experiment on convolution kernels of various sizes in the MSTCN module is performed to validate their effect on the classification result of the algorithm. The results are shown in Figure 9. The classification accuracy of the algorithm is better when the size differences of adjacent convolution kernels are two, and accuracy decreases as size differences increase.

## 5. Discussion

In MI-BCI research, MI paradigms of limb force were designed by mostly adopting hand clenching imagination tasks [14,15,16]. They were the MI paradigms of force under a unilateral upper-limb static state. In the brain-controlled rehabilitation robot system, these MI paradigms were difficult to be applied to the dynamic force interaction process between the robot and patient. To solve the above problem, we designed the MI paradigm of force under a unilateral upper-limb dynamic state based on the movement of wiping a table in human daily life. This paradigm introduced three levels of force into the same dynamic movement of the right upper limb. It increased the number of MI thinking categories with different levels of force under a unilateral upper-limb dynamic state. The ERD features of MI-EEG signals were analyzed in Section 4.1. The mean time–frequency maps at C3 and C4 channels showed that the three-level-force MI-EEG signals presented ERD features in both the alpha and beta bands, and the ERD features became more apparent with increasing force level of imagination. However, the feature differences were slight among single-trial MI-EEG signals at different levels of force. In addition, the mean power brain topography maps in two frequency bands (13–14 Hz and 24–28 Hz) showed that the brain’s contralateral sensorimotor area, which corresponded to three-level-force MI under the right upper-limb dynamic state, was activated, and the activated area became wider as the force level of imagination increased. Therefore, the space features of MI-EEG signals among different levels of force not only overlapped but were different.

Aiming at the above ERD features of MI-EEG signals, the MSTCN-AM recognition algorithm was proposed as a kind of deep neural network. We designed the MSTCN module with three small-size convolution kernels to extract fine-grained time–frequency domain features of different dimensions from the preprocessed single-trial MI-EEG signals. The feature information was richer in this way. On this basis, the spatial convolution module was used to learn the area differences of space domain features among different-level-force MI-EEG signals. In addition, we adopted the attention module using channel attention mechanism to weight the time–frequency–space domain features dynamically to improve the sensitivity of the algorithm and reduce feature redundancy. The results in Table 2 showed that the proposed MSTCN-AM algorithm achieved a classification accuracy of 86.4 ± 14.0% for recognition of three-level-force MI-EEG signals. The classification performance of this algorithm was better than that of other baseline algorithms. However, for subject S2, the classification accuracy was substantially lower than that of other subjects. The confusion matrix in Figure 7 showed that the accuracy of subject S2 was low for MI-EEG signals at each force level. In the MI experiment, all subjects were required to imagine three levels of force according to the difficulty of wiping the table, and the wiping direction was an alternating front and back cycle. In the process of imagination, subject S2 may not have been able to sufficiently imagine three levels of force because of brain tiredness, which had a negative effect on quality of three-level-force MI-EEG signals, resulting in low accuracy.

As a kind of traditional algorithm, OVR-CSP+SVM [32,33] requires manual extraction of EEG features. It only considers the space domain features but ignores the time–frequency domain features. As a result, limited feature learning directly affects the classification performance of the algorithm. For deep learning algorithms, Shallow ConvNet [19], Deep ConvNet [19], and EEGNet [24] are all end-to-end networks that automatically learn the time–frequency–space domain features and classify them. Therefore, manual extraction of EEG features is avoided and classification performance is improved. However, the single-scale TCN is employed in these algorithms so that the time–frequency domain features cannot be sufficiently explored, and classification performance is still limited. What is more, SCNN-BiLSTM [35] is a hybrid deep learning algorithm that combines CNN, LSTM, and attention mechanism. It learns the time–frequency–space domain features of EEG signals effectively, and the problem of insufficient feature learning is solved. However, this algorithm requires a large amount of training data, and its classification result is limited by the fitting effect of the network model on the EEG data. Overall, the above algorithms have their own limitations for feature recognition of different-level-force MI-EEG signals. In the above deep learning algorithms, a combination of multiple convolution layers is mostly used to extract features progressively. Although the combination of multiple convolution layers can learn the potential features in the time–frequency–space domain of MI-EEG signals, it cannot learn the internal correlation of features. In the MSTCN-AM algorithm we designed, it combines the MSTCN module, spatial convolution module, and attention module. The MSTCN module and spatial convolution module are used to sufficiently learn the time–frequency–space domain features of MI-EEG signals. The attention module is used to further learn the correlation among features of different feature channels. The results in Table 2 showed that the classification performance of the MSTCN-AM algorithm was better. This illustrates that our algorithm can effectively learn the distinguishable time–frequency–space domain features of MI-EEG signals among different levels of force. Furthermore, we compared the fitting performance of our algorithm and the EEGNet algorithm for MI-EEG data. The results showed that our algorithm had a higher convergence rate.

In addition, the combination of different-size convolution kernels determined the structure difference of the MSTCN module. Therefore, in order to verify its effect on the classification result of the MSTCN-AM algorithm, we carried out a combination experiment with multiple-size convolution kernels in Section 4.4. The results showed that the algorithm’s classification accuracy was better when the size differences of adjacent convolution kernels were two, while the accuracy decreased as the size differences continued to increase. The reason is that the feature differences are slight among single-trial MI-EEG signals at different levels of force. The larger the size differences among adjacent convolution kernels, the larger the receptive field of some convolution kernels. As a result, the fine-grained features of MI-EEG signals cannot be sufficiently extracted, thus affecting the classification performance of the MSTCN-AM algorithm.

In summary, compared with previous studies, we innovatively designed the MI paradigm with different levels of force under a unilateral upper-limb dynamic state. The MSTCN-AM algorithm was particularly proposed to recognize the ERD features of MI-EEG signals, which had better classification performance. However, the feature differences of EEG signals are slighter when MI thinking categories with different levels of force increase. The accuracy of our algorithm may be decreased. Therefore, in future research, we will optimize the network structure of our algorithm to solve the above problem. In our paradigm, although the force level of imagination changes among different single trials, it is fixed within one single trial. However, in the brain-controlled rehabilitation robot system, the force level of imagination changing within one single trial will better meet the demand for the natural and real-time interaction between the robot and patient. Therefore, in future research, we will focus on designing a novel MI paradigm of force sequence under the same dynamic movement of the unilateral upper limb. Aiming at the features of MI-EEG signals that change with force switching in a single trial, MI-EEG feature recognition algorithms with better accuracy and real-time performance will be deeply studied.

## 6. Conclusions

In this paper, we designed a MI paradigm, which increased the number of MI thinking categories with different levels of force under a unilateral upper-limb dynamic state. According to this paradigm, we analyzed the ERD features of MI-EEG signals and proposed the MSTCN-AM algorithm. In the time–frequency domain, aiming at the slight feature differences of single-trial MI-EEG signals among different levels of force, the fine-grained features of different dimensions were extracted by the MSTCN module with three small-size convolution kernels. In the space domain, the spatial convolution module was used to learn the area differences of space domain features. Finally, the attention module using channel attention mechanism was adopted to improve the sensitivity of the algorithm to the time–frequency–space domain features. The results showed that our algorithm’s classification accuracy for three-level-force MI-EEG data was 86.4 ± 14.0%. Compared with the baseline algorithms, our algorithm had higher classification accuracy, with significant differences and better fitting performance. The above research will lay the foundation for force control of a brain-controlled rehabilitation robot system in the movement direction.

## Figures and Tables

**Figure 1 entropy-25-00464-f001:**
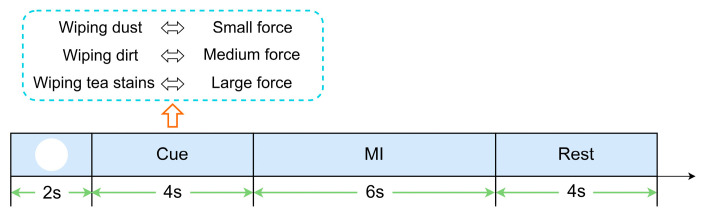
Single trial procedure for our experimental MI paradigm.

**Figure 2 entropy-25-00464-f002:**
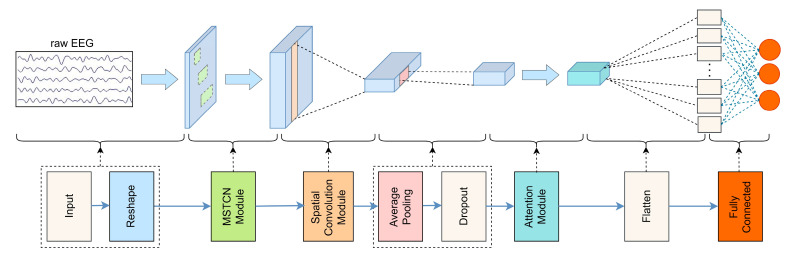
Network structure diagram. The MSTCN-AM algorithm consists of three major modules: the MSTCN module, the spatial convolution module, and the attention module.

**Figure 3 entropy-25-00464-f003:**
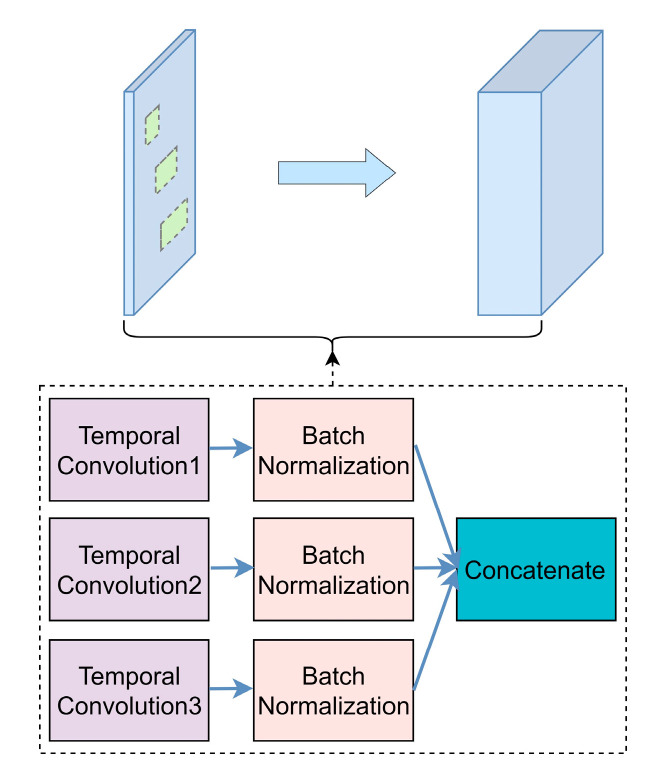
MSTCN module structure diagram.

**Figure 4 entropy-25-00464-f004:**
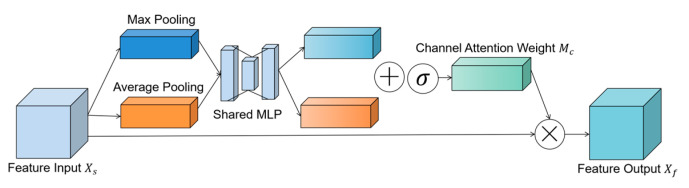
Attention module structure diagram. The “+”, “σ”, and “×” signs represent the operations of element-wise summation, activation function, and element-wise multiplication, respectively.

**Figure 5 entropy-25-00464-f005:**
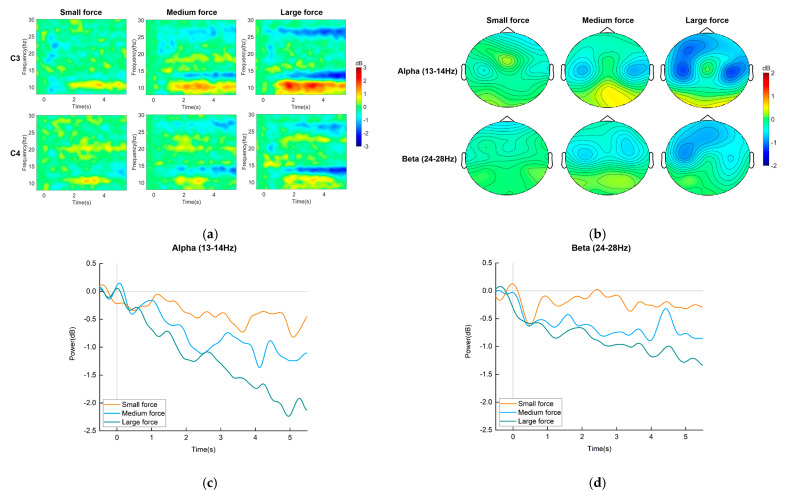
MI-EEG feature analysis results. (**a**) Mean time–frequency maps at C3 and C4 channels. A power decrease (blue) indicates an ERD during imagination of movement. (**b**) Mean power brain topography maps in the alpha and beta bands. “Blue” indicates ERD, which reflects activation of brain hemispheres. (**c**) ERSP curves in the alpha band. (**d**) ERSP curves in the beta band. The curves show how power (in dB) changes with time (in s).

**Figure 6 entropy-25-00464-f006:**
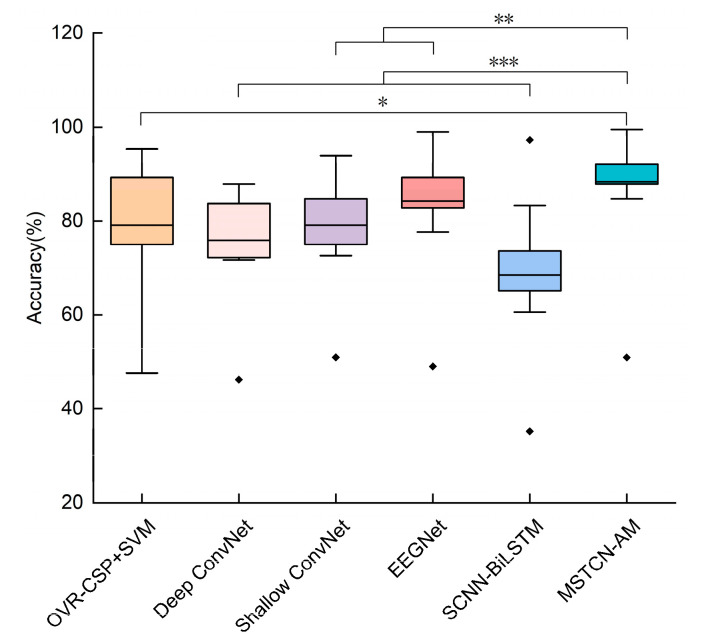
Box plot of classification accuracies of the MSTCN-AM algorithm and other baseline algorithms. The box plot shows the comparison of classification accuracy results in Table 2 and the comparison of statistical analysis results. Stars denote the statistically significant difference between two algorithms (where * denotes *p* < 0.05, ** denotes *p* < 0.01, and *** denotes *p* < 0.001).

**Figure 7 entropy-25-00464-f007:**
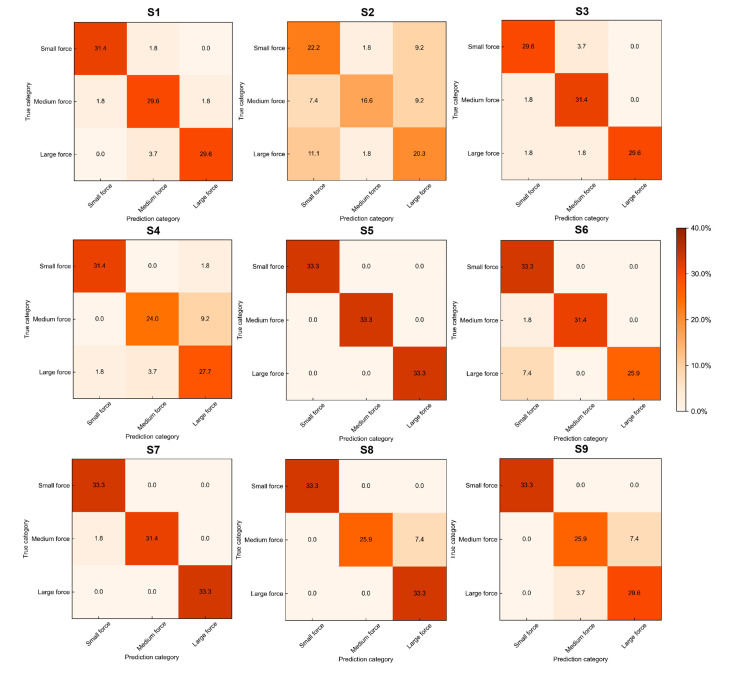
Confusion matrices of MI-EEG data test sets for all subjects (S1–S9). Each row and column of the confusion matrices represent the true category and prediction category of EEG signals, respectively.

**Figure 8 entropy-25-00464-f008:**
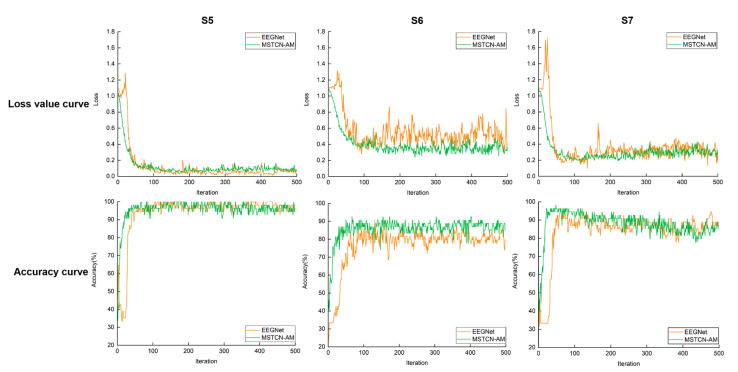
Loss value curves and accuracy curves of MI-EEG data test sets for three subjects (S5, S6, and S7). The orange line and green line represent the EEGNet algorithm and the MSTCN-AM algorithm, respectively. The curves show how the loss values or accuracy values change with 500 iterations. In the accuracy curves, the fewer iterations required for the algorithm to reach its maximum accuracy, the higher the convergence rate of the algorithm.

**Figure 9 entropy-25-00464-f009:**
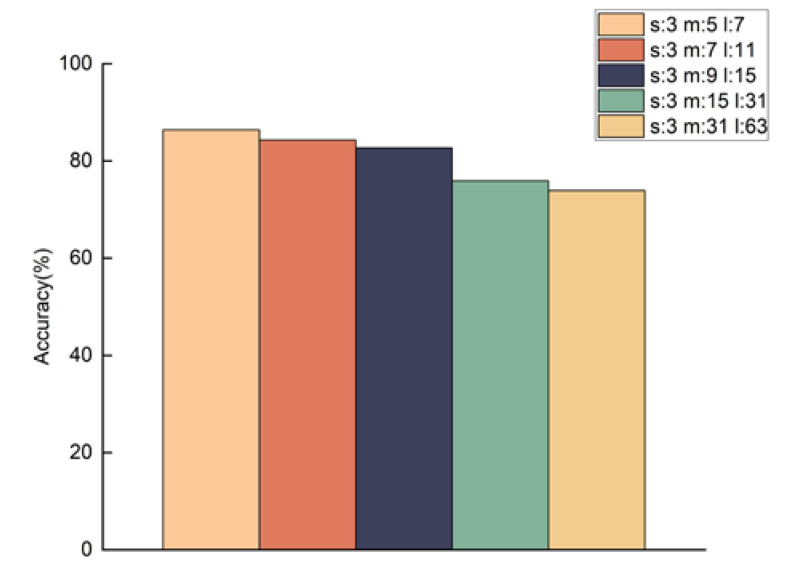
Classification accuracies of the MSTCN-AM algorithm with different combinations of three convolution kernels’ sizes. The letters s, m, and l represent the sizes of three convolution kernels in the MSTCN module, respectively.

**Table 1 entropy-25-00464-t001:** Network structure parameters. Parameters N and M are the number of electrodes and time points of EEG data, respectively. Parameter *p* is the dropout rate.

Module	Type	Layer	Parameter	Output	Activation
	Input	Input		(N, M)	
	Reshape	Reshape		(1, N, M)	
MSTCN	Convolution	Conv1	(1, 3)	(8, N, M)	Linear
	Batch Normalization	BN1		(8, N, M)	
	Convolution	Conv2	(1, 5)	(8, N, M)	Linear
	Batch Normalization	BN2		(8, N, M)	
	Convolution	Conv3	(1, 7)	(8, N, M)	Linear
	Batch Normalization	BN3		(8, N, M)	
	Concatenate	Concatenate		(24, N, M)	
Spatial Convolution	Convolution	Conv4	(N, 1)	(48, 1, M)	Linear
	Batch Normalization	BN4		(48, 1, M)	
	Activation	Activation		(48, 1, M)	ELU
	Average Pooling	Pool	(1, 64)	(48, 1, M/64)	
	Dropout	Drop	*p* = 0.5	(48, 1, M/64)	
Attention	Attention	Attention		(48, 1, M/64)	
	Flatten	Flatten		48 * M/64	
	Fully Connected	FC		3	Softmax

**Table 2 entropy-25-00464-t002:** Classification accuracy comparison for different algorithms.

	OVR-CSP +SVM	Deep ConvNet	Shallow ConvNet	EEGNet	SCNN- BiLSTM	MSTCN- AM
S1	75.0	84.7	85.6	83.3	72.2	88.4
S2	47.6	46.2	50.9	49.0	35.2	50.9
S3	89.3	80.5	80.0	89.3	73.6	89.8
S4	66.2	75.9	75.0	77.7	60.6	84.7
S5	95.3	87.9	93.9	99.0	97.2	99.5
S6	79.1	75.0	79.1	82.8	68.5	92.1
S7	75.0	83.7	84.7	90.7	83.3	96.2
S8	91.2	72.2	72.6	84.2	66.2	87.9
S9	80.5	71.7	77.3	85.1	65.2	88.4
Mean ± Std	77.6 ± 14.5	75.3 ± 12.3	77.6 ± 11.8	82.3 ± 13.8	69.1 ± 16.8	86.4 ± 14.0

## Data Availability

The data presented in this research are available on request from the corresponding author.

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
