# Peer review of "A Multi-Scale Temporal Convolutional Network with Attention Mechanism for Force Level Classification during Motor Imagery of Unilateral Upper-Limb Movements"

_entropy, 2023, doi:10.3390/e25030464_

Round 1

Reviewer 1 Report

I have reviewed this manuscript the overall contents of this manuscript is not well organized to give a clear overview of this work. I have suggested some comments about this work are as the following:

Comments to the Authors:

1.     Authors should write clearly abstract including results with numerical values in accuracy of all algorithms.

2.     In results section, author should revise the figure captions from Figure 1 and Figure 3. Explain clearly about the main findings of the figures in each captions.

3.      My suggestion is that the authors should write discussion section clearly in more details like how and why this MSTCN-AM algorithm with MI is important than previous BCI studies with MI. The current form of discussion is very short. Authors can referee the related article: “Ko L-W, Chikara RK, Lee Y-C, Lin W-C. Exploration of User’s Mental State Changes during Performing Brain–Computer Interface. Sensors. 2020; 20(11):3169.”

4.     The authors should write some limitations of MSTCN-AM algorithm in BCI system with MI.

Reviewer 2 Report

This manuscript proposed a novel framework of multi-scale temporal convolutional network with attention mechanism to recognize three different force levels in motor imagery brain computer interface tasks. Overall, the presentation of methods and results is clear, and the discussion is adequate. Below are my concerns.

Title. I think a title like “A Multi-Scale Temporal Convolutional Network with Attention Mechanism for Force Level Classification during Motor Imagery of Unilateral Upper Limb Movements” is a better fit.

Introduction. Need some more explanations to show why to involve attention mechanism in algorithm design, like why attention mechanism can dynamically weigh the input features.

Page 4. The title of 3.1, use “feature” instead of “characteristics”

Results. Besides the overall classification accuracy, please provide a confusion matrix to show the accuracy of each class.

Page 6, formula (4) for ELU function is not necessary, just cite.

Page 10, Table 2. Why the testing accuracy with S2 is substantially lower than other subjects for all methods? Need to provide some explanations.
